# Facing flood disaster: A cluster randomized trial assessing communities' knowledge, skills and preparedness utilizing a health model intervention

Mohd Tariq Mhd Noor[1], Hayati Kadir Shahar[1,2]*, Mohd Rafee Baharudin[1], Sharifah Norkhadijah Syed Ismail[1], Rosliza Abdul Manaf[1], Salmiah Md Said[1], Jamilah Ahmad[3], Sri Ganesh Muthiah[1]

1 Department of Community Health, Faculty of Medicine and Health Sciences, Universiti Putra Malaysia, Serdang, Selangor, Malaysia, 2 Malaysian Research Institute of Ageing (MyAgeing), Universiti Putra Malaysia, Serdang, Selangor, Malaysia, 3 School of Communication, Universiti Sains Malaysia, Malaysia, Malaysia

* hayatik@upm.edu.my

**Data Availability Statement:** All relevant data cannot be made available to the public due to

## Abstract

Floods occur when a body of water overflows and submerges normally dry terrain. Tropical cyclones or tsunamis cause flooding. Health and safety are jeopardized during a flood. As a result, proactive flood mitigation measures are required. This study aimed to increase flood disaster preparedness among Selangor communities in Malaysia by implementing a Health Belief Model-Based Intervention (HEBI). Selangor's six districts were involved in a single-blinded cluster randomized controlled trial Community-wide implementation of a Health Belief Model-Based Intervention (HEBI). A self-administered questionnaire was used. The intervention group received a HEBI module, while the control group received a health talk on non-communicable disease. The baseline variables were compared. Immediate and six-month post-intervention impacts on outcome indicators were assessed. 284 responses with a 100% response rate. At the baseline, there were no significant differences in ethnicity, monthly household income, or past disaster experience between groups (p>0.05). There were significant differences between-group for intervention on knowledge, skills, preparedness (p<0.001), Perceived Benefit Score (p = 0.02), Perceived Barrier Score (p = 0.03), and Cues to Action (p = 0.04). GEE analysis showed receiving the HEBI module had effectively improved knowledge, skills, preparedness, Perceived Benefit Score, Perceived Barrier Score, and Cues to Action in the intervention group after controlling the covariate. Finally, community flood preparedness ensured that every crisis decision had the least impact on humans. The HEBI module improved community flood preparedness by increasing knowledge, skill, preparedness, perceived benefit, perceived barrier, and action cues. As a result, the community should be aware of this module.

**Clinical trial registration:** The trial registry name is Thai Clinical Trials Registry, trial number TCTR20200202002.

ethical restrictions. Data can be provided upon request to the Universiti Putra Malaysia Ethical Committee for Human Subjects Research. Protecting the privacy and confidentiality of research participants is obligatory. Data can be requested from the Deputy Dean (Research and Internationalization) of the Faculty of Medicine and Health Sciences at Universiti Putra Malaysia, 43400 Serdang, Selangor, Malaysia. Office telephone number: +60397692501 Email: upa_medic@upm. edu.my.

**Funding:** The funding body is the Department of Research Excellence Division of the Ministry of Higher Education Malaysia. The funding is external and not industry-funded, and the included relevant documentation was uploaded as an additional file. The study had undergone full external peer review before the research grant was awarded. The award number is JPT.S(BPKI)2000/09/01/046(51), and the award recipient is Associate Professor Dr. Hayati Binti Kadir @Shahar. The funders had no role in study design, data collection and analysis, decision to publish, or preparation of the manuscript.

**Competing interests:** The authors have declared that no competing interests exist.

**Abbreviations:** ANOVA, Analysis of Variance; HEBI, Health Education Based Intervention; HBM, Health Belief Model; GEE, Generalized Estimating Equations; NADMA, Malaysian National Disaster Preparedness Agency; MOH, Ministry of Health.

## Introduction

Floods are the most common type of natural disaster, occurring when a body of water over-flows and submerges ordinarily dry land. Flooding is frequently caused by heavy rain, rapid snowmelt, or a storm surge from a tropical cyclone or tsunami along the coast. Floods can wreak havoc, claiming lives and destroying personal property as well as critical public health infrastructure. Between 1998 and 2017, floods affected over 2 billion people worldwide. Floods are hazardous for people who live in floodplains or non-resistant structures or lack flood warning systems and awareness [1].

Flood disasters have become more common in recent decades, especially the devastating storms and floods that many attribute to climate change. In the 2000s, around 150 major floods were recorded worldwide, more than tripling the number recorded in the 1980s. The number of deaths caused by natural disasters, particularly floods, can vary significantly yearly; some have very few deaths before a significant flood event claims many lives. Over the last decade, natural disasters have killed approximately 60,000 people per year on average world-wide. It is responsible for 0.1% of all deaths worldwide [2]. Thousands of people have died in South Asia, and millions have been affected by the worst floods in decades. Some villages in the area have been thoroughly washed away, while others have been submerged beneath a watery coffin. Even so, the floodwaters have cut off more, making it nearly impossible to deliver much-needed and life-saving aid. For example, one-third of Bangladesh's land area is now submerged underwater. Buildings in Mumbai, India's financial capital, are collapsing, killing those who cannot flee. Crop and farmland damage has significantly reduced the amount of food available. Tens of thousands of people will have to rely on food distribution from aid organizations for the foreseeable future. Evacuees from Chor Asadia village in the Brahmaputra need food [3].

Flooding is Malaysia's most devastating natural disaster. Only 0.014% of the water on Earth is salt water, with the rest (97%) found in lakes, rivers, underground, and the atmosphere [4]. Floods in Malaysia are classified into two types by the Malaysian Drainage and Irrigation Department: flash floods and monsoon floods. Floods are the most common natural disaster in Malaysia, prone to seasonal monsoon floods, flash floods, and tidal floods. Floods have had severe consequences for people, affecting livelihoods, destroying property and infrastructure, and claiming lives [5]. The annual flood damage is approximately $274 million, and lives are lost yearly [5]. It is responsible for most of the frequent and significant damage. Floods are also responsible for many human deaths, disease outbreaks, property and crop damage, and other losses. Floods are the most critical contributor, accounting for 62.5% of fatalities and causing 62.5% of all economic damage [6]. The year-end floods and downpours in 2014 were the worst in the country's history, affecting over 500,000 people. The cost of infrastructure damage alone has been estimated to be USD 670 million (RM 2.851 billion) [6].

Flood disaster vulnerability refers to the characteristics and circumstances of a community, system, or asset that make it vulnerable to flooding. There are four types of vulnerability: social vulnerability, physical vulnerability, economic vulnerability, and environmental vulnerability. People living in flood-prone areas, the poor, children, the elderly, women, refugees, unregistered migrants, internally displaced people, and trafficked people are among the many vulnerable groups in Malaysia [7]. As a result, a flood preparedness plan is essential. However, people cannot participate in local disaster preparedness plans due to a lack of knowledge [8]. Members of the community felt they lacked access to information and didn't know how to create local disaster preparedness plans. They were completely unaware of the significance of these

plans. Furthermore, the community could not identify local government officials responsible for specific disaster preparedness issues to inform them of the challenges in involving community members in local disaster preparedness or the lack of resources to launch community-level disaster preparedness initiatives [9]. To improve Malaysian community disaster preparedness, some issues must be addressed. Changing how a community or society views disaster is one of the most difficult challenges. Malaysians are not accustomed to major disasters because they are rarely exposed to them. Most of the time, people believe that major disasters will not affect their homes or villages. In communities that have not been directly affected by disasters or have never experienced one, the perception is worse. In the event of a disaster, this mindset impacts the community's response, preparedness, and level of concern.

Vulnerable citizens require disaster preparedness training. There are numerous approaches to educating marginalized people, none of which is superior to the others. People who have received a good education will be better able to defend themselves and others. Disaster preparedness necessitates the development and implementation of robust training programmes in this regard [10]. Flood disaster preparedness provides a platform for developing effective, realistic, and coordinated planning, which reduces duplication of efforts and increases the overall effectiveness of flood disaster preparedness and response efforts by National Societies, households, and community members [11]. Flood disaster preparedness activities combined with risk reduction measures can help prevent disasters while saving as many lives and livelihoods as possible during any disaster situation, allowing the affected population to return to normalcy quickly. A continuous and integrated process resulting from a wide range of risk reduction activities and resources rather than a single sectoral action. Contributions from a wide range of sectors are required, including training and logistics, health care, recovery, livelihood, and institutional development [12–14].

In Malaysia, the current educational intervention provides disaster preparedness information without incorporating behaviour change theory to influence behaviour changes and address associated factors. The Health Belief Model is used as the theory in most intervention studies in other countries, with the Ecological Theory, Social Cognitive Theory, and KAP Theory being used in a few others [8, 15–19]. Hence, the Health Belief Model (HBM) is applied to disaster preparedness efforts mainly focuses on human behaviour [20] and increase their ability to cope with hazard consequences. Individual attitudes and beliefs were used to develop the Health Belief Model. Four main pillars represent the perceived threat and net benefits. It has a sense of susceptibility, severity, benefits, and barriers to overcome. This model was created to evaluate people's willingness to act. Disaster preparedness, on the other hand, can benefit individuals as well as communities. Community resilience is critical, and it can assist residents in making plans. By encouraging the community to take action to prepare for disasters and recover from them, the community resilience preparedness approach promotes a strong community facing disasters and recovering from them [21]. This HEBI module in this study has been thoroughly developed, based on HBM theory, to demonstrate that a person's belief in a personal health or disease threat, combined with confidence in the effectiveness of the prescribed health behaviour or action, predicts someone's likelihood to adopt the behaviour.

Hence, this study will also contribute to the body of knowledge to better understand interventions that improve the community's knowledge, skills, and preparedness. If this intervention is effective, this research will be added to the existing disaster preparedness programme. It can help agencies like the National Disaster Management Agency (NADMA) and the Ministry of Health Malaysia (MOH) prepare individuals and communities for flood disasters. HEBI can also help with the 2030 Agenda for Sustainable Development, which recognises and reaffirms the critical need to reduce disaster risk. There are specific opportunities to achieve SDGs through disaster risk reduction, in addition to direct references to the Third United Nations

Conference on Disaster Risk Reduction (Sendai Framework) outcomes. For example, it could be reducing the poor's disaster exposure and vulnerability, or building intervention readiness [22].

## Research objectives

The overall goal is to develop, implement, and assess a Health Belief Model-based intervention (HEBI) on knowledge, skills, and preparedness in Selangor communities, Malaysia.

## Material and methods

### Setting and participants

This study was a two-arm, parallel, single-blind study that ran from September 2019 to April 2021. First, Selangor state's six districts were chosen based on similarities in their flood-related histories. Following that, the districts were ranked based on the frequency of floods each year and the severity of the disaster, as determined by data from Malaysia's Department of Irrigation and Drainage. Furthermore, the sampling population consisted of communities from Selangor's district that met the inclusion and exclusion criteria: Malaysian citizens aged 18 and up, with illiteracy and physical disability serving as exclusion criteria. Moreover, the COVID-19 pandemic impacted the study during the recruitment of participants and data collection. Numerous appointments with participants had to be rescheduled because of Movement Order Control. The report's overall completion was also impacted. Hence, the research team wrote to the funder requesting an extension of the study duration.

### Intervention

The intervention programme aimed to improve disaster knowledge, skills, and preparedness constructs from HBM among the community in the flood-prone area of Selangor. The intervention was delivered throughout two sessions. First, face-to-face sessions included an educational presentation, a group discussion, a video field visit, and mapping and revising community vulnerability and flood disaster plans. In the second stage, the respondent can be reached personally by phone and via WhatsApp follow-up. Following the recruitment, the face-to-face sessions included an educational presentation and a group discussion. A total of ten to fifteen people were polled. The researcher led the educational talk and group discussion in the hall or meeting room. A PowerPoint presentation and a video demonstration were shown to the participants. The intervention began with a two-hour health education talk. The content of the health education talk includes an introduction to flood disasters, statistics on flood disasters in the world and Malaysia, an introduction to Malaysian flood disaster management, community preparation before a flood disaster, and the consequences of an unprepared flood disaster.

Respondents watched a video produced by the Malaysian Department of Irrigation and Drainage and the Malaysian Ministry of Health. Following that, a small group discussion (between 3 and 5 participants per session) was held to address any questions or concerns. Participants presented a list of scenarios and situations, and problems and solutions were discussed. Participants were encouraged to raise any issues or concerns they had about disaster preparedness in their community, as well as any other related matters.

Previous interventional studies used the Health Belief Model, Social Cognitive Theory, and Ecological Theory constructs. According to the literature review, many interventions utilizing HBM effectively improved community members' flood disaster preparedness [8, 17, 18, 23].

As a result, this intervention module was developed based on previous interventional studies, using the HBM as the construct.

Some of the HBM constructs used in the intervention module were perceived susceptibility, perceived benefit, perceived severity, perceived barrier, self-efficacy, and cues to action. Furthermore, respondents' knowledge, skills, and preparedness were among the factors covered by the constructs. Table 1 explains the working framework for using HBM's behavioural change techniques to improve flood disaster preparedness among the respondents in this study.

## Outcomes

The primary outcomes are increased knowledge, skills, and preparedness scores. The secondary outcomes are the Health Belief Model constructs (susceptibility, severity, benefit, barrier, cues to action, and self-efficacy scores).

## Sample size

The sample size (N) for this study was calculated by comparing the populations of two groups using Lemeshow and Lwanga's formula (1991) [24]. We calculated the sample size by considering the intracluster correlation coefficient, the number of events, the expected effect, and the power of the study. We assumed an intracluster correlation of = 0.005, and the final sample size was estimated to be 284 for both the intervention and control groups. Given the small cluster size and rare outcome, these calculations included a 10% design effect, which was expected to be negligible.

## Randomization, allocation, and blinding

The participants were randomly assigned to one of two groups: intervention or control. As a result, three districts were given to the intervention group and three others to the control group. To prevent communication between the intervention and control groups and minimize contamination, participants were recruited based on the districts they represent. The group

**Table 1. The application of the behavioural change techniques of HBM in educational intervention to flood disaster preparedness.**

| Stage | HEBI components | Theoretical model constructs | Method |
|---|---|---|---|
| Knowledge | Group education (health talk and group discussion) Surrogate experimental learning (Brochure, flier, poster, video) | Perceived Benefit Perceived Barrier Self-Efficiency Perceived susceptibility | General Knowledge about disaster preparedness was assessed, and all material will be disseminated during the intervention. |
| | | | Perceived barrier and perceived benefit were assessed during the discussion about the disaster preparedness barrier and benefit of preparedness. |
| | | | Every individual was assessed by baseline questionnaire before intervention programmed. |
| | | | The intention of the preparedness was assessed. |
| Skills | Interactive learning with the community (brainstorming) | Cues to action | Information related to advantage and disadvantage of preparedness |
| | | | Information about disaster preparedness was explaining. |
| | | | Information on HEBI intervention how to prepare individuals for disaster. |
| Preparedness and implementation | First Aid demonstration | Cues to action | Preparation of disaster kit was assessed before and after intervention Available of emergency contact and plan of action after intervention |
| | Video experimental learning with the group mapping the vulnerability area at community level, and revisit at community level) | | |

assignment was kept a secret from both the participants and the trainee personnel (research assistants) using a single-blind method.

## Quality control

Maturation effects can occur rapidly, within a few hours or days. Participants' responses during data collection may vary depending on whether they are in a good or negative mood at the time. Excessive fatigue, boredom, hunger, and inattention can all influence response. A participant may have received little sleep before to data collection for a project, resulting in fatigue, or may be busy with other duties, resulting in inattention. These participant-based factors can be challenging to manage and reduce the internal validity of findings. There may also be selection bias in determining which organizations will be included in the intervention or control groups.

The content was validated by four public health experts and one medical expert in terms of questionnaire validity. Each item's content validity ratio was calculated, and each item received a minimum value of 0.5. The questionnaire has been translated into *Bahasa Melayu*, the researcher's native language, by four public health experts and one medical expert. For face validity, participants first evaluated the questionnaire's comprehensiveness and clarity regarding what it intends to measure. Second, whether the questionnaire is simple, easy to understand, contains inappropriate, redundant, or missing items, and how likely it will address the research objectives. Third, whether the questionnaire's flow, arrangement, and wording are reliable.

## Statistical methods

SPSS 25.0 was used for all analyses. The association between intervention groups were determined using simple linear regression, and the predictors were determined using multiple linear regression, as applicable. ANOVA with repeated measures evaluated baseline, Time 1 (immediate post-intervention), and Time 2 (at 6 months). On the other hand, the Generalized Estimating Equations assessed the statistical differences on multiple continuous dependent variables by independent variable while controlling for covariates.

## Ethical consideration

Ethical clearance obtained from Ethics Committee for Research Involving Human Subjects Universiti Putra Malaysia (Jawatankuasa Etika Universiti Untuk Penyelidikan Melibatkan Manusia (JKEUPM) with JKE approval number UPM/TNCPI/RMC/JKEUPM/1.4.18.2 (JKEUPM). Besides that, written informed consent from the respondents has obtained before the study.

## Results

### Response rate

Six district clusters in Selangor were screened for eligibility. Two hundred eighty-four people from six districts agreed to take part in the trial. Six districts were successfully randomized into the intervention (3 districts) and control (3 districts) groups, with 142 respondents from both groups. The baseline response rate, immediate post-intervention response rate, and 6-month post-intervention response rate were 100% (284 respondents). All 284 respondents completed all three follow-up points. The final research flow chart based on the CONSORT statement is depicted in Fig 1 [25].

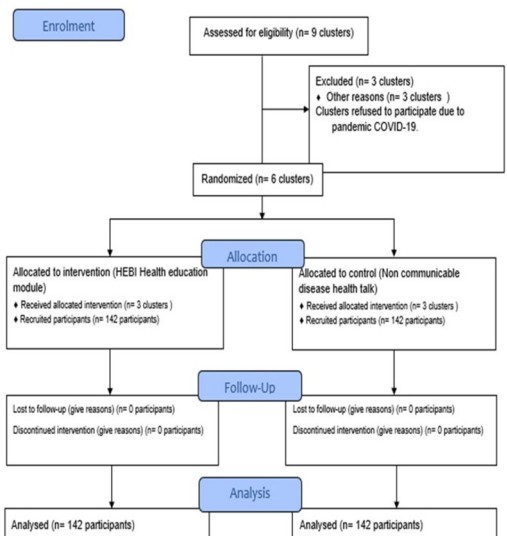

**Fig 1. Flow chart diagram of the study using CONSORT statement.**

## Sociodemographic and socioeconomic characteristics of the participants

The sociodemographic characteristics of the participants are shown in Table 2. The median ages of the control and intervention groups were 40 years old (IQR = 21) and 46 years old (IQR = 23), respectively. Most participants in the intervention group were male, while most participants in the control group were female. Most participants in both the intervention and control groups were employed. Table 2 also shows Chi-square analyses of associations between sociodemographic, socioeconomic, and personal characteristics and the intervention and control groups. Ethnicity, monthly household income, and personal factors were insignificant in both the intervention and control groups.

Table 3 compares the intervention and control groups' baseline results for knowledge, skills, preparedness, susceptibility, severity, benefit, barrier, cues to action, and self-efficacy. There was no significant difference between the intervention and control groups at the baseline level in all variables except skills and perceived barrier score.

## Effectiveness of the primary outcome and secondary outcome

This section described the analysis objective five, which compared the effectiveness of HEBI in terms of knowledge, skills, preparedness, and disaster perception between the intervention and control groups at baseline, immediate, and six months after controlling for covariates and was analyzed using Generalized Estimating Equations (GEE).

**Effectiveness of HEBI in increasing knowledge, skill, and preparedness for disaster.** A GEE analysis was performed to see if the Knowledge, Skill, and Preparedness scores differed between the comparison groups at the immediate and six-month follow-ups. According to the model information, the model is a binomial legitimate model with knowledge, skill, and preparedness as the dependent variable. It considers the between-group effect (participant and cluster of districts) and the within-group effect (time). The unstructured correlation matrix is used as the working correlation matrix because it produces the best fit model. A total of 6 clusters and 284 participants were analyzed as correlated data for the two-time points of immediate follow-up and six months follow-up. The correlation matrix dimension was two. There were 284 participants in this study, and there was no attrition, so the response rate was 100%.

**Table 2. Baseline comparison of sociodemographic characteristics of participants between the intervention and control group.**

| Characteristics | Group median (IQR) | | | df | p-value |
|---|---|---|---|---|---|
| | Control | Intervention | | | |
| Age (Years) | 40(21) | 46(23) | | | |
| | Group n (%) | | | df | p-value |
| | Control | Intervention | Total | | |
| **Sociodemographic** | | | | | |
| **Gender** | | | | | |
| Male | 60(42.3) | 111(78.2) | 171(60.3) | 1 | <0.001* |
| Female | 82(57.7) | 31(21.8) | 113(39.8) | | |
| **Marital Status** | | | | | |
| Single | 20(14.1) | 19(13.4) | 39(13.7) | 2 | 0.01* |
| *Married | 102(71.8) | 120(84.5) | 222(78.2) | | |
| Divorcee/Widow | 20(14.1) | 3(2.1) | 23(8.1) | | |
| **Ethnicity** | | | | | |
| Malay | 132(93) | 130(91.5) | 262(92.3) | 3 | 0.771 |
| Chinese | 4(2.8) | 5(3.5) | 9(3.2) | | |
| Indian | 6(4.2) | 6(4.2) | 12(4.2) | | |
| Others | 0(0) | 1(0.7) | 1(0.4) | | |
| **Education** | | | | | |
| Not formal education | 14(9.9) | 0(0) | 14(4.9) | 3 | <0.001* |
| Primary | 38(26.8) | 32(22.5) | 70(24.6) | | |
| Secondary | 68(47.9) | 89(62.7) | 157(55.3) | | |
| Degree and above | 22(15.5) | 21(14.8) | 43(15.1) | | |
| **Employment Status** | | | | | |
| Employed | 74(52.1) | 114(80.3) | 188(66.2) | 1 | <0.001* |
| Unemployed | 68(47.9) | 28(19.7) | 96(33.8) | | |
| **Monthly household income** | | | | | |
| <RM4000 | 122(85.9) | 118(83.1) | 240(84.5) | 2 | 0.232 |
| RM4000-RM8500 | 18(12.7) | 24(16.9) | 42(14.8) | | |
| >RM8500 | 2(1.4) | 0(0) | 2(0.7) | | |
| **Car ownership** | | | | | |
| Yes | 102(71.8) | 124(87.9) | 226(79.9) | 1 | 0.01* |
| No | 40(28.2) | 17(12.1) | 57(20.1) | | |
| **Personal Characteristics** | | | | | |
| **Past disaster experience** | | | | | |
| Yes | 96(67.6) | 82(57.7) | 178(62.7) | 1 | 0.086 |
| No | 46(32.4) | 60(42.3) | 106(37.3) | | |
| **Knowledge about disaster preparedness** | | | | | |
| Yes | 44(31) | 63(44.4) | 107(37.7) | 1 | 0.06 |
| No | 98(69) | 79(55.6) | 177(62.3) | | |

*Significant at p<0.05

The average age of the participants was 44.65 (±13.76SD). Both arms were equal among the 142 participants in the control and intervention groups. Gender, marital status, education, employment status, and car ownership were all adjusted covariates.

Table 4 shows the effectiveness of HEBI on Knowledge between intervention and control group. The analysis showed a significant difference observed in knowledge between the trial

**Table 3. Baseline result of knowledge score, skills score, preparedness score, susceptibility, severity, benefit, barrier, cues to action, and self-efficacy score between intervention and control group.**

| Baseline Characteristics | Median (IQR) | | U | t | p-value |
|---|---|---|---|---|---|
| | Intervention | Control | | | |
| Knowledge Score[a] | 7(3) | 7(3) | 9494 | | 0.384 |
| Skills Score[b] | 33.48(6.33) [c] | 31.83(6.48) [c] | | -2.33 | 0.021* |
| Preparedness Score[a] | 3(6) | 1(4) | 9024 | | 0.244 |
| Perceived Susceptibility Score[a] | 22(5) | 21(5) | 9322 | | 0.270 |
| Perceived Severity Score[a] | 10(4) | 11(4) | 9631 | | 0.505 |
| Perceived Benefit Score[b] | 14.2(5.15)[c] | 14.2(4.10)[c] | | -1.045 | 0.297 |
| Perceived Barrier Score[a] | 45(15) | 47(11) | 8321 | | 0.014* |
| Cues to Action Score[b] | 13.9(5.64)[c] | 14.2(4.22)[c] | | -1.765 | 0.079 |
| Self-efficacy Score[b] | 13.8(5.37)[c] | 14.2(4.31)[c] | | -0.509 | 0.611 |

*Significant at p<0.05

[a]Mann-Whitney U test

[b]Independent Sample t-test

[c]Mean (SD)

groups ($\beta = 0.67$, CI = 1.61, 2.37, p<0.05). On the contrary, no significant difference was observed in knowledge scores on timepoints at immediate follow-up ($\beta = 0.28$, CI = 0.79, 1.55, p<0.53). Otherwise, a significant effect was observed within the timepoint at 6-months ($\beta = -0.74$, CI = 1.65, 2.64, p<0.05) after adjusting the covariates. The knowledge of the respondents showed to be increased up to 0.74 times higher compared to the baseline illustrated by Fig 2. There was significant direct interaction between the intervention group and both timepoints at 6-month follow-up ($\beta = 1.26$, CI = 2.26, 5.52, p<0.01).

Table 5 shows the effectiveness of HEBI on skill between intervention and control group. The analysis showed a significant difference observed in skill between the trial groups ($\beta = 4.72$, CI = 49.29, 253.03, p<0.05). Furthermore, there was a significant difference in skill scores at the timepoints of immediate follow-up and 6-month follow-up. Respondent skills were up

**Table 4. Effectiveness of HEBI on Knowledge between intervention and control group.**

| Variable | B[c] | SE | Wald | 95% CI | | p-value |
|---|---|---|---|---|---|---|
| | | | | Lower | Upper | |
| **Trial Group** | | | | | | |
| Control[a] | | | | | | |
| Intervention | 0.67 | 0.99 | 45.02 | 1.61 | 2.37 | <0.001* |
| **Timepoints** | | | | | | |
| Baseline[a] | | | | | | |
| immediate follow-up | 0.28 | 0.15 | 0.77 | 0.79 | 1.55 | 0.53 |
| 6-month follow-up | 0.74 | 0.12 | 37.94 | 1.65 | 2.64 | <0.001* |
| **Group*Time** | | | | | | |
| Intervention at immediate follow-up | 0.18 | 0.20 | 0.83 | 0.81 | 1.79 | 0.36 |
| Intervention at 6-month follow-up | 1.26 | 0.23 | 30.79 | 2.26 | 5.52 | <0.001* |

[a]Reference group

[b]Pooled estimate after multiple imputations

[c]Intercept B coefficient of 6.812 for this model

*Significant at p<0.05

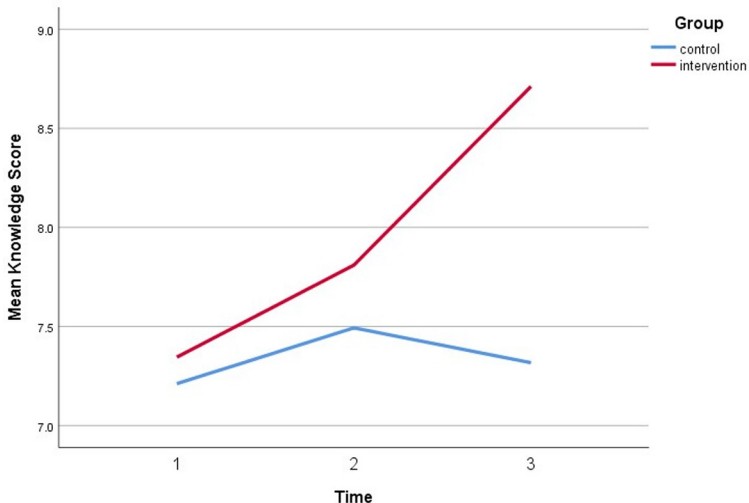

**Fig 2. A plot of knowledge scores among respondents, showing the interaction between groups and time.**

to four times higher than the baseline at immediate time points and two times higher at the 6-month follow-up shown in Fig 3. Otherwise, after controlling for covariates, a significant effect was observed within the 6-month time point. As a result, the intervention group and both time points of follow-up had significant direct interaction.

Table 6 shows the effectiveness of HEBI on preparedness between intervention and control group. There was a significant difference observed in skill between the trial groups (β = 1.16, CI = 1.61, 6.39, p<0.05). Those at 6-month post-intervention were one time likelier to increase preparedness score than baseline after adjusting covariate (β = 0.71, CI = 0.28, 0.85, p<0.05). Then those who were intervention arm were five times more likely to increase preparedness score compared to the control arm after adjusting for covariates (gender, marital status, education, employment status, and car ownership) (β = 5.26, CI = 80.92, 464.08, p<0.05) demonstrated in Fig 4.

**Table 5. Effectiveness of HEBI on Skill between intervention and control group.**

| Variable | B[c] | SE | Wald | 95% CI | | p-value |
|---|---|---|---|---|---|---|
| | | | | Lower | Upper | |
| **Trial Group** | 4.716 | .4173 | 127.687 | 49.288 | 253.032 | <0.001* |
| Control[a] | | | | | | |
| Intervention | | | | | | |
| **Timepoints** | | | | | | |
| Baseline[a] | | | | | | |
| immediate follow-up | 4.25 | 0.83 | 25.92 | 13.68 | 361.73 | <0.001* |
| 6-month follow-up | 2.05 | 0.71 | 8.41 | 0.03 | 0.51 | 0.04* |
| **Group*Time** | | | | | | |
| Intervention at immediate follow-up | -2.02 | 0.96 | 4.40 | 0.02 | 0.87 | 0.036* |
| Intervention at 6-month follow-up | 9.13 | 0.99 | 84.79 | 1325.46 | 64704.15 | <0.001* |

[a]Reference group

[b]Pooled estimate after multiple imputations

[c]Intercept B coefficient of 24.39 for this model

*Significant at p<0.05

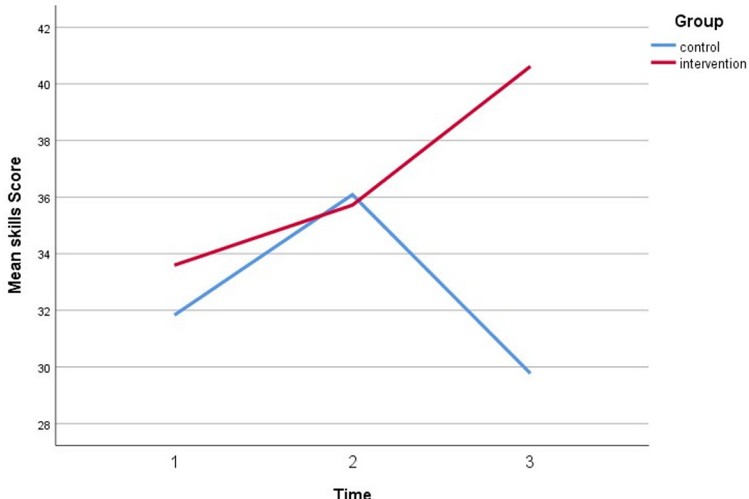

**Fig 3. A plot of skills scores among respondents, showing an interaction between groups and time.**

**Effectiveness of HEBI in increasing disaster perception disaster (secondary outcome).**
Tables 7–9 demonstrate the efficacy of disaster perception in this study. Except for the perceived barrier score, all measured perceptions show a significant direct interaction between the intervention group and the 6-month timepoint follow-up. Perceived susceptibility increased by one-point, perceived severity increased by one-point, perceived benefit increased by 0.5 points, and cues to action increased by 1.16 points. Furthermore, after controlling for covariates, self-efficacy increased by 0.93 points compared to the baseline (gender, marital status, education, employment status, and car ownership).

Simultaneously, all perceived assessments show a significant direct interaction between the intervention group and both timepoint follow-ups. The perceived susceptibility score increased by three points, the perceived severity score increased by three points, the perceived

**Table 6. Effectiveness of HEBI on preparedness between intervention and control group.**

| Variable | B[c] | SE | Wald | 95% CI | | p-value |
|---|---|---|---|---|---|---|
| | | | | Lower | Upper | |
| **Trial Group** | 1.16 | 0.35 | 10.90 | 1.61 | 6.39 | 0.001* |
| Control[a] | | | | | | |
| Intervention | | | | | | |
| **Timepoints** | | | | | | |
| Baseline[a] | | | | | | |
| immediate follow-up | 1.95 | 0.29 | 43.91 | 3.94 | 12.46 | <0.001* |
| 6-month follow-up | 0.71 | 0.28 | 6.30 | 0.28 | 0.85 | 0.01 |
| **Group*Time** | | | | | | |
| Intervention at immediate follow-up | 0.71 | 0.38 | 3.35 | 0.22 | 1.05 | 0.06 |
| Intervention at 6-month follow-up | 5.26 | 0.44 | 139.74 | 80.92 | 464.08 | <0.001 |

[a]Reference group

[b]Pooled estimate after multiple imputations

[c]Intercept B coefficient of 0.768 for this model

*Significant at p<0.05

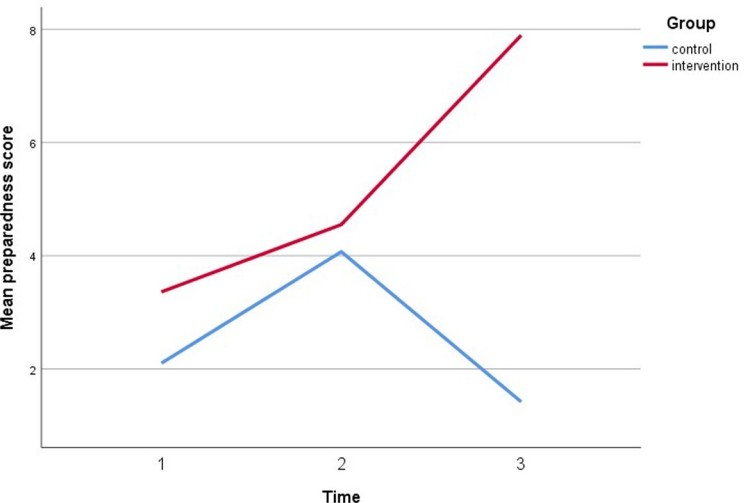

**Fig 4. A plot of preparedness scores among respondents, showing an interaction between groups and time.**

benefit score increased by five points, and the cues to action score increased by five points. Furthermore, after controlling for covariates, self-efficacy increased by 6 points compared to

**Table 7. Effectiveness of HEBI on perceived susceptibility and perceived severity score.**

| Perceived Item | Variable | B[c] | SE | Wald | 95% CI | | p-value |
|---|---|---|---|---|---|---|---|
| | | | | | Lower | Upper | |
| **Perceived Susceptibility score** | **Trial Group** | | | | | | |
| | Control | | | | | | |
| | Intervention | 0.22 | 0.51 | 0.19 | 1.246 | 0.46 | 0.66 |
| | **Timepoints** | | | | | | |
| | Baseline | | | | | | |
| | immediate follow-up | 0.96 | 0.53 | 3.32 | 0.93 | 7.34 | 0.06 |
| | 6-month follow-up | 0.556 | 0.42 | 1.79 | 0.25 | 1.29 | 0.18 |
| | **Group*Time** | | | | | | |
| | Intervention at immediate follow-up | 0.411 | 0.65 | 0.39 | 0.414 | 5.49 | 0.533 |
| | Intervention at 6-month follow-up | 3.73 | 0.60 | 38.21 | 12.82 | 136.95 | <0.001* |
| | Intercept B coefficient | 19.81 | | | | | |
| **Perceived Severity score** | **Trial Group** | | | | | | |
| | Control | | | | | | |
| | Intervention | 0.549 | 0.44 | 1.55 | 0.73 | 4.10 | 0.21 |
| | **Timepoints** | | | | | | |
| | Baseline | | | | | | |
| | immediate follow-up | 0.61 | 0.35 | 0.73 | 2.99 | 4.10 | 0.08 |
| | 6-month follow-up | 0.17 | 0.38 | 0.21 | 0.39 | 1.77 | 0.05* |
| | **Group*Time** | | | | | | |
| | Intervention at immediate follow-up | 1.66 | 0.53 | 10.04 | 0.064 | 0.52 | 0.002* |
| | Intervention at 6-month follow-up | 3.09 | 0.54 | 32.51 | 7.62 | 63.86 | <0.001* |
| | Intercept B coefficient | 11.46 | | | | | |

[a]Reference group

*Significant at p<0.05

**Table 8. Effectiveness of HEBI on perceived benefit and perceived barrier score.**

| Perceived Item | Variable | B[c] | SE | Wald | 95% CI | | p-value |
|---|---|---|---|---|---|---|---|
| | | | | | Lower | Upper | |
| **Perceived Benefit Score** | **Trial Group** | | | | | | |
| | Control[a] | | | | | | |
| | Intervention | 0.62 | 0.55 | 1.27 | 0.63 | 5.52 | 0.02* |
| | **Timepoints** | | | | | | |
| | Baseline[a] | | | | | | |
| | immediate follow-up | 2.20 | 0.53 | 14.54 | 2.68 | 21.55 | <0.001* |
| | 6-month follow-up | 0.49 | 0.46 | 1.14 | 0.24 | 1.52 | 0.02* |
| | **Group*Time** | | | | | | |
| | Intervention at immediate follow-up | -0.33 | 0.73 | 0.21 | 0.17 | 3.02 | <0.001* |
| | Intervention at 6-month follow-up | 5.02 | 0.72 | 49.08 | 37.32 | 620.94 | <0.001* |
| | Intercept B coefficient | 19.62 | | | | | |
| **Perceived Barrier Score** | **Trial Group** | | | | | | |
| | Control[a] | | | | | | |
| | Intervention | 0.62 | 0.55 | 1.27 | 0.63 | 5.55 | 0.026* |
| | **Timepoints** | | | | | | |
| | Baseline[a] | | | | | | |
| | immediate follow-up | 2.02 | 0.53 | 14.54 | 2.68 | 21.55 | <0.001* |
| | 6-month follow-up | -0.49 | 0.46 | 1.14 | 0.24 | 1.51 | 0.28 |
| | **Group*Time** | | | | | | |
| | Intervention at immediate follow-up | -0.33 | 0.73 | 0.21 | 0.17 | 3.02 | 0.65 |
| | Intervention at 6-month follow-up | 5.03 | 0.72 | 49.08 | 37.32 | 620.94 | <0.001* |
| | Intercept B coefficient | 19.62 | | | | | |

[a]Reference group

*Significant at p<0.05

baseline (gender, marital status, education, employment status, and car ownership). Figs 5–10 depicted respondents' interaction with groups and over time.

## Discussion

The HEBI Module's primary goal is to improve disaster preparedness knowledge, skills, and readiness. During the immediate follow-up and six months, each group's knowledge, skills, and preparedness increased. The intervention group saw a greater increase than the control group. The increase in knowledge, skill, and preparedness lasted six months within each group. The increase was statistically significant within each group. The six-month follow-up period coincided with a Pandemic Covid-19 Movement Control Order. This HEBI's efficacy was comparable to that of the systematic review on the optimal dose for intervention.

For HEBI of this study, a total of 60 articles were identified by the electronic search strategy after removed the duplicates study. Twenty articles were removed after reviewing the title and abstract. Twenty articles were retrieved and screened for eligibility. For secondary screening, 15 articles that fulfilled the inclusion and exclusion criteria were reviewed, and only nine articles were included in the systematic review after full articles were reviewed depicted in Fig 11.

For optimal dose intervention of this study, the electronic search strategy identified a total of 60 articles after removing the duplicated studies. Twenty articles were removed after reviewing the title and abstract. Twenty articled were retrieved and screened for eligibility. For

**Table 9. Effectiveness of HEBI on cues to action and self-efficacy score.**

| Perceived Item | Variable | B<sup>c</sup> | SE | Wald | 95% CI | | p-value |
|---|---|---|---|---|---|---|---|
| | | | | | Lower | Upper | |
| **Cues to Action** | **Trial Group** | | | | | | |
| | Control[a] | | | | | | |
| | Intervention | 0.977 | 0.48 | 4.12 | 1.03 | 6.82 | 0.04* |
| | **Timepoints** | | | | | | |
| | Baseline[a] | | | | | | |
| | immediate follow-up | 2.51 | 0.48 | 27.22 | 4.78 | 31.42 | <0.001* |
| | 6-month follow-up | 1.16 | 0.52 | 4.97 | 0.113 | 0.87 | 0.026* |
| | **Group*Time** | | | | | | |
| | Intervention at immediate follow-up | -1.27 | 0.61 | 4.37 | 0.08 | 0.92 | 0.03* |
| | Intervention at 6-month follow-up | 5.07 | 0.74 | 46.54 | 37.06 | 681.96 | <0.001* |
| | Intercept B coefficient | 13.37 | | | | | |
| **Self-efficacy** | **Trial Group** | | | | | | |
| | Control[a] | | | | | | |
| | Intervention | 0.39 | 0.60 | 0.42 | 0.46 | 4.82 | 0.515 |
| | **Timepoints** | | | | | | |
| | Baseline[a] | | | | | | |
| | immediate follow-up | 2.27 | 0.58 | 15.19 | 3.1 | 30.58 | <0.001* |
| | 6-month follow-up | 0.93 | 0.52 | 3.32 | 0.143 | 1.07 | 0.06 |
| | **Group*Time** | | | | | | |
| | Intervention at immediate follow-up | -1.74 | 0.82 | 4.41 | 0.03 | 0.88 | 0.036* |
| | Intervention at 6-month follow-up | 6.48 | 0.85 | 57.49 | 122.96 | 3522.26 | <0.001* |
| | Intercept B coefficient | 29.53 | | | | | |

[a]Reference group

*Significant at $p < 0.05$

secondary screening, a total of 15 articles that met the inclusion and exclusion criteria were reviewed, and only eight articles were included in the systematic review after full articles were reviewed shown in Fig 12.

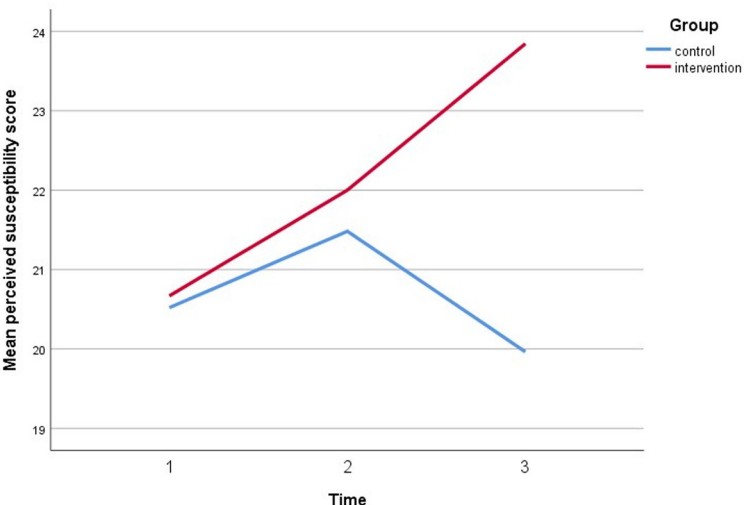

**Fig 5. The plot of perceived susceptibility scores among respondents showing an interaction.**

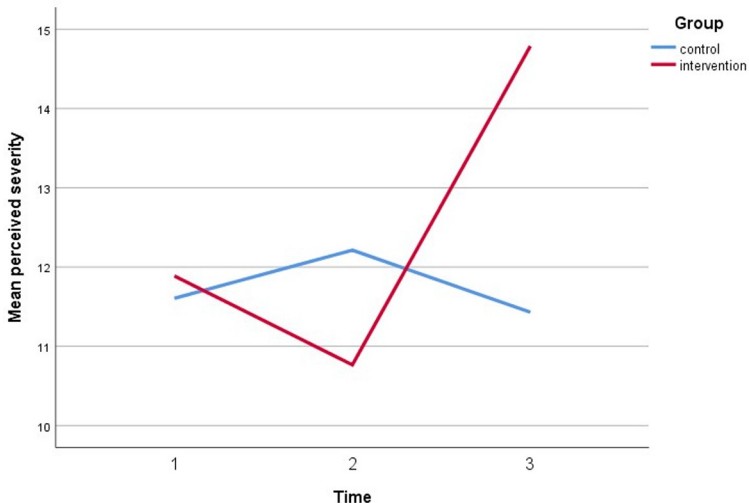

**Fig 6. A plot of perceived severity scores among respondents, showing an interaction between groups and time.**

Five of the eight papers cited used intervention only once during the study, two used it twice, and one used it three times [8, 15, 18, 19, 26–28]. Many of these studies employed a single six-month intervention that was carried out concurrently. Between the evaluation and the report, there was a three-month lag. Most of the findings matched those of the Iranian study. Three provinces in Iran were studied, as was a study of urban and rural areas similar to that done in the United States, a study in a remote region of Golestan, and a study in Los Angeles [8, 16, 26]. All four of these studies demonstrated that there was still a significant level change for the items studied by the researcher three months after the intervention. Overall, the reviewed papers revealed that the optimum dose for each intervention trial, whether one, two, three, or more doses, had no significant effect on the final result. However, other factors may need to be investigated in the future to determine the effectiveness of a respondent's action. According to the research summarised here, an educational intervention programme with only one intervention dosage will prevent the population from receiving the optimal dose of

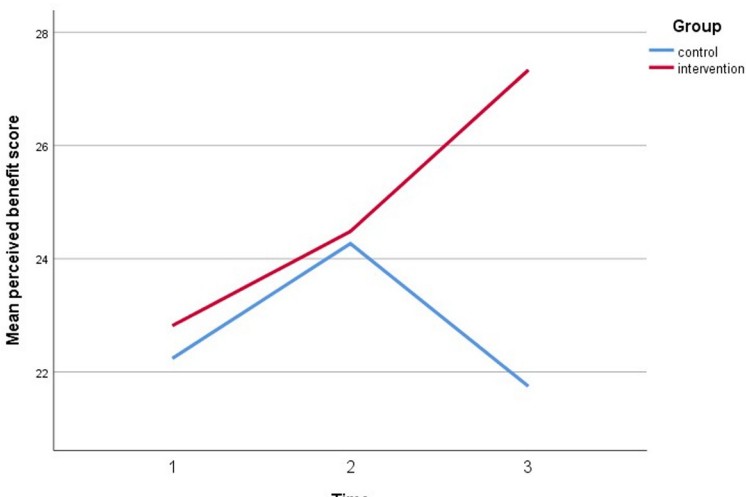

**Fig 7. A plot of perceived benefit scores among respondents, showing an interaction between groups and time.**

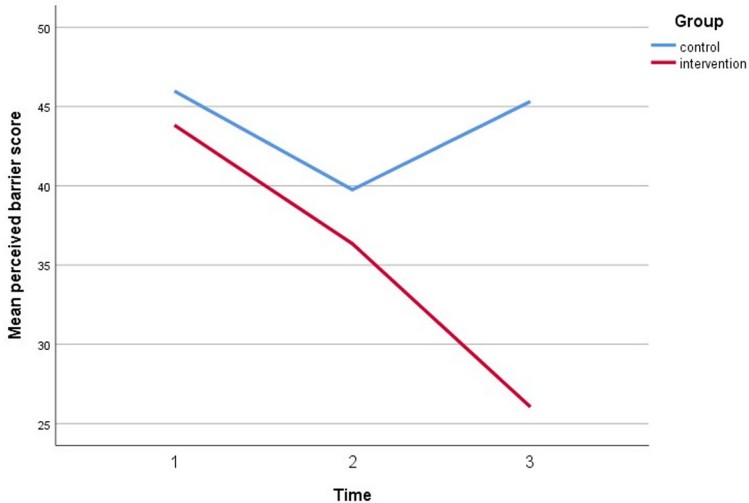

**Fig 8. A plot of perceived barrier scores among respondents, showing an interaction between groups and time.**

disaster preparedness intervention. As a result, even at the immediate follow-up or six months after the HEBI is given to the community, the HEBI has demonstrated efficacy. It agrees with previous research findings.

According to the research, if a group is given information based on the HBM theory, its preparedness in the face of flood disasters may change [29]. Furthermore, this study discovered that group roles (attitude and knowledge) play an important role in determining community disaster awareness. In the context of this study, a well-informed population in the studied community with the necessary skills and mindset leads to increased disaster awareness. According to the majority of respondents, there is a significant level of community attitude, knowledge, and disaster awareness [30]. In this case, many respondents believe that careful planning on community roles would increase community engagement in disaster awareness and response. Assume that appropriate community service programmes are prioritised. In

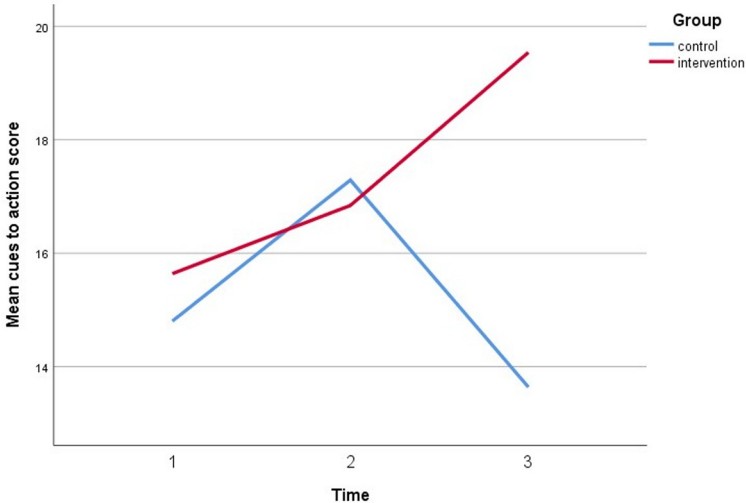

**Fig 9. A plot of cues to action scores among respondents, showing an interaction between groups and time.**

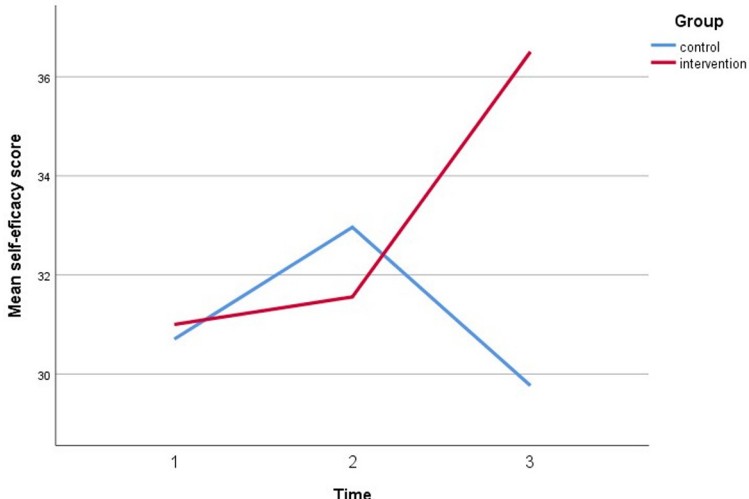

**Fig 10. A plot of self-efficacy scores among respondents, showing an interaction between groups and time.**

that case, they will assist community leaders in managing their communities on an individual and group basis, resulting in increased disaster awareness in the community. As a result, the studied group must implement full community roles (community attitude and knowledge) to ensure the studied community's long-term success in disaster awareness [30]. It implies that this HEBI Module employs relevant theories and concepts to ensure that the community is prepared in the event of a flood disaster.

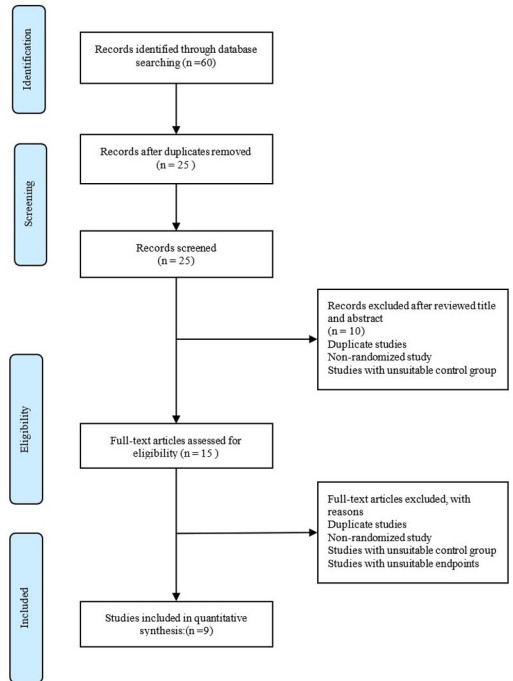

**Fig 11. PRISMA flow diagram of literature search for health education intervention disaster preparedness among community in improving disaster preparedness among community.**

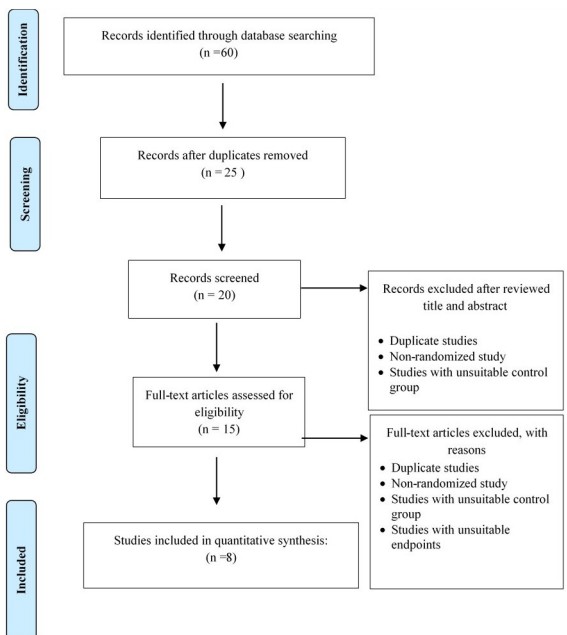

**Fig 12. PRISMA flow diagram of literature search for optimal dose of disaster preparedness intervention utilising health belief model theory.**

On the other hand, other studies advocate for the dissemination of information or education to improve community disaster preparedness [31]. When the group was adequately informed and prepared for the disaster, they could react and recover quickly.

The HEBI module was developed and implemented with care to ensure that any observation differences and current module implementation were taken into account. As a result, HEBI follows the principle of the Health Beliefs Model. Previously, systematic research was conducted in the methodology chapter. It summed up clear evidence that implementing an educational intervention programme or module based on the Health Belief Model theory can help communities prepare for disasters. The studies consistently demonstrated remarkable efficacy in raising community awareness, skill, and preparedness, prompting the development of various methodological approaches. This systematic review discovered that disaster preparedness interventions based on the Health Belief Model (HBM) theory improved the study's quality, particularly the methodology. It should be noted, however, that each problem is equally significant. We must choose an educational-based intervention described in the theory above to implement an intervention model that results in meaningful improvement. The HBM hypothesis is frequently used because the Health Belief Model is a theoretical model that can guide health promotion and disease prevention initiatives [32]. It is used to explain and forecast how people's health habits change. It is one of the most popular models for researching health-related behaviours. The previous systematic review identified interventional studies that use the HBM as the theoretical basis for intervention design [32]. For the past 40 years, the HBM has been used to develop behaviour modification approaches. However, 14 (78%) of the 18 studies examined showed significant changes in adherence, while 4 (22%) showed small overall results. Only six studies used the HBM completely, and five of them looked at health attitudes as a result [33].

Furthermore, when participants received the intervention of this HEBI module, the primary outcome, namely knowledge, skills, and preparedness, showed significant changes

compared to the baseline. Within-group disaster perception was as useful as those who did not receive HEBI intervention. At the same time, selecting the correct theory during module manufacturing is critical to delivering the appropriate and accurate intervention [34]. Significant differences in skill can be found between the intervention and control groups. Previous research has shown that basic skills such as early warning, first aid, triage, logistics and communication, search and rescue, and team organisation are required to assist victims during a disaster's emergency response. More importantly, simple first-aid procedures such as controlling bleeding, treating shock, and stabilising fractures are critical for assisting most victims of natural disasters with injuries [35].

Compared to the baseline, there was a significant difference in the intervention group that received HEBI for the preparedness items. Disaster management training, i.e., knowledge and skills, improve the technical skills of disaster relief workers and volunteers and prepares them to face the disaster as a whole, according to [36]. Other studies conducted in Bangladesh show that if participants' knowledge and skills change significantly, so will their preparedness to face flood disasters [37]. According to social scientists specialising in social perception, people's perceptions or understanding of natural disasters are socially constructed. It means that any understanding of how people perceive natural hazards should take into account the context in which natural hazards are encountered [38]. As a result, disaster risk perception is primarily of a universal and theoretical nature, and it serves as a means of achieving risk understanding and flood disaster preparedness. As a result, the Sendai Framework places a premium on increasing risk comprehension plans to achieve community resilience [39].

A Lagos study found that the majority of respondents were aware of their vulnerability to floods. Respondents who have previously experienced flooding, on the other hand, are more aware of the risks associated with flooding. It also identifies flood risk anxiety as an important predictor of flood risk preparedness. Furthermore, it demonstrates that flood disaster perception was higher among those concerned about a flood risk disaster and lower among those who were not. Finally, it demonstrates a close but positive relationship between flood risk awareness and level of preparedness, lending support to the idea that flood risk awareness leads to flooding risk preparedness. Based on preliminary findings, the paper advocates for improved flood risk awareness at the community level as a critical strategy for increasing disaster risk preparedness and community resilience to flooding risk disasters in Lagos [40].

According to the findings of a Turkish study, psychological first aid training provided to nursing students improved all phases of disaster preparedness and general self-efficacy perceptions. As a result, it is suggested that psychological first aid training be expanded. Furthermore, the students' scale scores decreased slightly in follow-up measurements [41]. According to studies conducted in the Philippines, Filipinos who believe climate change is directly affecting their households are more likely to take precautions to prepare for disasters ($p < 0.01$). Furthermore, Filipinos who believe climate change has directly harmed them are more likely to plan for disasters, make plans, and take concrete precautions, such as home modifications ($p < 0.01$) [42].

Another study discovered a link between people's perceptions of global climate change and their awareness of the risk of local climate disasters. The public's perception of global climate change is critical in raising public awareness of the dangers of local climate-related disasters in Cartago. The study identified two additional actors who have the potential to improve community awareness of local climate-related disaster risk. One way to become more aware of a community's social hazards is to share minor disaster experiences with larger groups or cities. Another study discovered a link between people's perceptions of global climate change and their awareness of the risk of local climate disasters. The public's perception of global climate change is critical in raising public awareness of the dangers of local climate-related disasters in

Cartago. The study identified two additional actors who have the potential to improve community awareness of local climate-related disaster risk. One way to become more aware of a community's social hazards is to share minor disaster experiences with larger groups or cities. Despite increased community awareness of local climate-related disaster risk, the study discovered that communities had implemented few disaster-related initiatives for a variety of reasons. The study identified three criteria for increasing future disaster response: improving daily living conditions, providing learning opportunities for communities to incorporate disaster risk reduction into daily life, and reawakening a desire to help neighbours improve their quality of life. These factors are critical for strengthening local government disaster management capabilities in the face of increasing climate-related disasters. These variables should be considered in regional development planning [43].

The first step in obtaining baseline data about their community's ability to respond to flood disasters is to assess their perception of their preparedness, knowledge, and skills for flood disaster preparedness. Before developing goals and objectives for flood disaster training and education initiatives, effective flood disaster training and education initiatives rely on input from the target population. The study's findings identified critical areas of flood disaster preparedness, training, and education to meet the needs of community settings for efficient and timely flood disaster response. According to the study's findings, and as evidenced by previous studies, when the community's perception of flood disaster preparedness improves, so does their knowledge, skill, and preparedness for disasters. It can also assist planners and coordinators in developing emergency plans and researching future guidelines [44].

## Conclusion

This study indicated the HEBI module enhanced community flood preparedness by increasing knowledge, skill, preparedness, perceived benefit, perceived barrier, and action cues which it showed the research objectives have been achieved.

This research was meticulously planned to produce high-quality evidence. A randomised controlled trial (RCT) is one of the gold standard pieces of evidence for determining the effect of the central intervention if it is carried out correctly. The primary goal is to eliminate or reduce significant source bias. A randomised cluster trial's primary goal is to solve the contamination problem. According to [45], the individual RCT trialist should be used because the RCT cluster will increase the sample size and cause recruitment bias. Individual RCTs can maintain contamination levels of up to 30%. In terms of practicality, action should always be grouped. It is customary in this community to regard the community as social behaviour, which leads to a high level of contamination when moving from one location to another. It occurs when people in the control arm are exposed to the procedure. Each of Selangor's six districts is represented in this study. Different courses were available in each district. There was no district programme during the research period.

As a result, those in the control group cannot be considered influenced. According to additional disaster preparedness studies, individuals in the HEBI community improved in the primary outcome of knowledge, skills, and preparedness. Simultaneously, there was an increase in disaster awareness (secondary outcome). The cluster design mitigated the effects of contamination. A sub-analysis of those not chosen from the sample revealed no significant differences in minimising differential selection. To avoid bias in recruitment, the recruiting process was completed before sampling.

Recruitment bias is another issue in cluster RCTs. In most cluster studies, conducting a double-blind RCT is extremely difficult. Respondents are recruited and interventions are delivered by the same people. When recruiters in the two arms act differently and are aware of

the offered intervention, there is a risk of bias. Recruitment can be done before random sampling to address this issue. The purpose of the preliminary questionnaire was to determine the level of disaster preparedness in the community and to identify potential participants. Following that, districts were randomly assigned, and all qualified participants were randomly sampled, ensuring that the recruitment processes were not influenced by intervention group exposure.

Future researchers, public health practises, and policymakers are all mentioned in this study's recommendations. First, some theory elements and additional variables, such as the level of Protective Behaviour, are discussed [33]. Second, a qualitative analysis should be included in future studies to gather feedback and develop the material. Focus group discussions, for example, will highlight components of the intervention that will be beneficial and components that need to be improved using a qualitative design. Finally, the study's follow-up period should be extended to one year to monitor results and avoid pandemic COVID-19 and the fasting month. The procedure should be repeated more frequently and frequently to ensure a long-term impact, especially after one month. Monthly intervals are feasible and realistic to implement. More testing should be carried out to see how different time intervals affect the results.

Based on public health results, the HEBI module intervention had a reasonable rate of participation and promising results. This HEBI module has been thoroughly developed, based on HBM theory, to demonstrate that a person's belief in a personal health or disease threat, combined with confidence in the effectiveness of the prescribed health behaviour or action, predicts the likelihood of someone adopting the behaviour. Other agencies, government agencies, and states are included in this intervention. The HEBI module intervention can also be used in the national disaster preparedness programme as part of the counselling module, with other Ministerial examples in the Malaysia Ministry of Health. The intervention will encourage the community to prepare and try again until they succeed with enough resources. This HEBI modular can also be combined with other policymakers' modules, such as the Malaysian National Disaster Preparedness Agency (NADMA) or the Malaysian Ministry of Health (MOH). It must ensure that every perception is identified, and that the community has the necessary knowledge, skills, and is prepared for flood disasters.

## Supporting information

**S1 Checklist. PRISMA 2020 for abstracts checklist.**
(PDF)

**S2 Checklist. PRISMA 2020 checklist.**
(PDF)

**S1 Dataset.**
(SAV)

## Acknowledgments

We appreciate the guidance and help given by the Department of Community Health, Faculty of Medicine, Universiti Putra Malaysia, during the writing process.

## Author Contributions

**Conceptualization:** Hayati Kadir Shahar, Rosliza Abdul Manaf, Salmiah Md Said, Jamilah Ahmad, Sri Ganesh Muthiah.

**Formal analysis:** Mohd Tariq Mhd Noor, Mohd Rafee Baharudin, Sharifah Norkhadijah Syed Ismail.

**Investigation:** Mohd Rafee Baharudin, Sharifah Norkhadijah Syed Ismail.

**Methodology:** Mohd Tariq Mhd Noor, Mohd Rafee Baharudin, Sharifah Norkhadijah Syed Ismail, Salmiah Md Said, Jamilah Ahmad, Sri Ganesh Muthiah.

**Supervision:** Hayati Kadir Shahar, Rosliza Abdul Manaf, Salmiah Md Said, Jamilah Ahmad, Sri Ganesh Muthiah.

**Validation:** Hayati Kadir Shahar, Rosliza Abdul Manaf, Salmiah Md Said, Jamilah Ahmad, Sri Ganesh Muthiah.

**Visualization:** Hayati Kadir Shahar, Rosliza Abdul Manaf, Salmiah Md Said, Jamilah Ahmad, Sri Ganesh Muthiah.

**Writing – original draft:** Mohd Tariq Mhd Noor, Mohd Rafee Baharudin, Sharifah Norkhadijah Syed Ismail.

**Writing – review & editing:** Hayati Kadir Shahar, Rosliza Abdul Manaf, Salmiah Md Said, Jamilah Ahmad, Sri Ganesh Muthiah.

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
