## [Decision Letter · Decision Letter 0]

21 Dec 2021

PONE-D-21-36005Facing Flood Disaster: Assessing Communities' Knowledge, Skills and Preparedness utilizing a Health Model Intervention.PLOS ONE

Dear Dr. Kadir Shahar,

Thank you for submitting your manuscript to PLOS ONE. After careful consideration, we feel that it has merit but does not fully meet PLOS ONE’s publication criteria as it currently stands. Therefore, we invite you to submit a revised version of the manuscript that addresses the points raised during the review process.

We look forward to receiving your revised manuscript.

Kind regards,

Beverley J Shea, Ph.D

Academic Editor

PLOS ONE

Journal Requirements:

[We appreciate the guidance and help given by the Department of Community Health, Faculty of Medicine, Universiti Putra Malaysia, during the writing process. Sponsor Malaysian Research University Network grant with grant number MRUN Grant JPT.S(BPKI)2000/09/01/046(51), All Project Members from Universiti Putra Malaysia (UPM), Universiti Teknologi Malaysia (UTM), Universiti Kebangsaan Malaysia (UKM), Universiti Sains Malaysia (USM) and Research Management Centre, UPM.]

[The funding body is the Department of Research Excellence Division of the Ministry of Higher Education Malaysia. The funding is external and not industry-funded, and the included relevant documentation was uploaded as an additional file. The study had undergone full external peer review before the research grant was awarded. The award number is JPT.S(BPKI)2000/09/01/046(51), and the award recipient is Associate Professor Dr. Hayati Binti Kadir @Shahar. 

The funders had no role in study design, data collection and analysis, decision to publish, or preparation of the manuscript.]

4. PLOS requires an ORCID iD for the corresponding author in Editorial Manager on papers submitted after December 6th, 2016. Please ensure that you have an ORCID iD and that it is validated in Editorial Manager. To do this, go to ‘Update my Information’ (in the upper left-hand corner of the main menu), and click on the Fetch/Validate link next to the ORCID field. This will take you to the ORCID site and allow you to create a new iD or authenticate a pre-existing iD in Editorial Manager. Please see the following video for instructions on linking an ORCID iD to your Editorial Manager account: https://www.youtube.com/watch?v=_xcclfuvtxQ.

6. Please ensure that you refer to Figures 2-4 in your text as, if accepted, production will need this reference to link the reader to the figure.

Reviewers' comments:

Reviewer's Responses to Questions

**Comments to the Author**

1. Is the manuscript technically sound, and do the data support the conclusions?

Reviewer #1: Yes

Reviewer #2: Yes

2. Has the statistical analysis been performed appropriately and rigorously? 

Reviewer #1: Yes

Reviewer #2: Yes

3. Have the authors made all data underlying the findings in their manuscript fully available?

Reviewer #1: Yes

Reviewer #2: Yes

4. Is the manuscript presented in an intelligible fashion and written in standard English?

Reviewer #1: Yes

Reviewer #2: Yes

5. Review Comments to the Author

Reviewer #1: The review claims that a Health Belief Model-based intervention that incorporates the behavior change theory to influence behaviour changes, and address associated factors, would improve Malaysia’s Selangor communities’ knowledge, skills, and preparedness for flood disasters.

This is a valid claim noting that The Health Belief Model is a widely applied theory for community-based intervention studies. It has also been found to improve flood disaster preparedness in various countries. The authors mention other possible theories but opt for ‘The Health Belief Model’ in disaster preparedness efforts. This decision is based on the existing gap in Malaysia’s current educational intervention, the model’s suitability in focusing on human behaviour and its consequent ability to manage hazard aftermaths.

All data collected and the subsequent analyses support the study claims. The cluster randomized controlled trial method was used to conduct the research.

The paper has followed the CONSORT guidelines but with the following shortcomings:

- The manuscript is not explicitly identified as a cluster randomized trial in the title.

- The source of funding is not given in the abstract.

- 3 of 9 eligible clusters refused to participate in the research due to the coronavirus disease (COVID-19). Exclusion of the 3 clusters, though captured in Figure 1, (a CONSORT flow diagram), is not explicitly recorded in the descriptive text. The authors state a 100% retention of the dataset at baseline, however, that ‘original’ dataset excludes an eligible 33.3%.

Considering the operative analytical sample, the methodology is technically sound and therefore able to allow reproduction of the experiments. This is based on the following:

- Participants were randomly assigned to a group.

- Both participants and trainee personnel were blinded.

- Questionnaires used were validated.

- There was low risk of contamination between the groups.

- There was no ongoing district program during the time of the research.

- There was 100% retention of the identified analytical sample following exclusion of the 3 above-mentioned clusters.

The experiments were conducted rigorously, and purpose outlined. The necessary adjustments for covariates were done and statistical analyses were performed and presented in detail. The data collected and analyzed support the conclusions that have been drawn.

The authors stated that the data underlying the findings is available without restriction, however, they also state that datasets used and analyzed during the study are available from the corresponding author on reasonable request. It is not clear whether that is a restriction to this review, given that there is no indication of whether the URLs/accession numbers/DOIs will be available only after acceptance of the manuscript for publication.

The manuscript is presented in an intelligible fashion and written in standard English. It can be accessible to non-specialists. However, the following considerations would ensure better organization and clarity across the manuscript:

- Page 5 - line 108 to 109: “Because Malaysians are rarely exposed to major disasters, they are accustomed to them” – The intended connotation of this sentence is unclear.

- Page 7-8: There is repetition between lines 163 and 174

- Page 12 - line 228: The final sample size given as 84. This does not tally with the 284 number referred to under ‘Results and Response Rates’ on line 282 for example. Further clarification is needed.

- Page 13 - line 249 to 256: The paragraph on maturation seems misplaced or not adequately explained.

- Page 10 - Table 1, under methods: “Perceived benefit and perceived benefit were assessed….” “Perceived benefit” is repeated. In addition, the first columns of Tables 7, 8 and 9 require some formatting.

- Figures 1 to 12 require complete labelling. Though they can still be correctly interpreted as presented, explicit labeling of both axis’ is suggested.

- Page 29 - Line 496 and 497: “This HEBI's efficacy was comparable to that of the systematic review in chapter 2 regarding the optimal dose for ….”. “Chapter 2” is not referenced, and it is unclear where to locate it in the manuscript.

- Line 552 and 553: “The previous systematic review identified interventional studies that use the HBM as the theoretical basis for intervention design.” It is not clear which ‘previous systematic review’ the authors are referring to.

- Page 37 - Line 554 to 556: “However, 14 (78%) of the 18 studies examined showed significant changes in adherence, while 7 (39%) showed small overall result.” There seems to be a possible error. If 14 out of 18 showed significant changes then perhaps it should read 4 (22%). Further clarification may be necessary.

In my opinion, this manuscript does not contain Dual Use Research of concern. Consequent to my search, the study also appears to present results of an original research.

A study protocol by the manuscript authors is available using the following link - https://doi.org/10.1186/s12889-021-11719-3. It is titled "A cluster-randomized trial study on effectiveness of health education based intervention (HEBI) in improving flood disaster preparedness among community in Selangor, Malaysia: a study protocol".

The following deviations from the protocol are noted:

- The manuscript makes no mention of data collection at 3 months post-intervention as is stated in the protocol’s abstract.

- The protocol indicates that the research will include 6 districts. There was no explicit statement regarding the three that refused to participate due to COVID-19 as stated in the manuscript’s flow chart.

Should the paper be considered unsuitable for publication in its present form, the authors should be encouraged to submit a revised version. I will make myself available to answer questions from the editors and for a re-review of the manuscript.

Thank you.

Reviewer #2: This is an interesting manuscript. I am proposing the following minor revisions:

- Lines 108-109: Grammar/Sentence structure: "Because Malaysians...to them". This sentence needs revision, it is confusing.

- Lines 164-165: Please describe exactly how COVID-19 impacted the study. This sentence makes it seem like major disruptions occurred in all aspects of the study. Additional information about how and how the study team mitigated these effects is required.

- Lines 168-Lines 174: Is this information a repeat of the above? Please revise.

- Section 3.2: I suggest that this section should begin at line 197 because it is not clear to me what the intervention actually is until this paragraph. The information about HBM (lines 181-196) can follow.

- Section 3.6 is a repeat of section 3.5? Please revise, this should not be the case.

- The paragraph starting at line 249 is not clear and requires revision. Specifically, lines 252-253.

- Line 339: "Table 4 shows..." What is 'arm'? If you are explaining that results were compared across intervention and control, this sentence needs revision to be more clear. It was not an outcome studied. Same for Line 362 and 384.

- Section 6.0 Conclusion: Should start with a short concise summary of the results and how the objective of the study was achieved.

- Lines 681-684: "This HEBI module... adopting the behaviour" is very important rationale and is not clear at the start of the manuscript. Please revise to include this information earlier on for the reader.

6. PLOS authors have the option to publish the peer review history of their article (what does this mean?). If published, this will include your full peer review and any attached files.

Reviewer #1: No

Reviewer #2: No

---

## [Author Response · Author response to Decision Letter 0]

6 Jun 2022

We Thank Editor and Reviewers. Below are our responses.

1. The ORCID ID has been authenticated in "update my information" section. ORCID 0000-0002-7841-5957

2. Figure 2-4 has been depicted in the text.

3. Anonymized Data set has been provided in the revision system.

4. We have uploaded the completed PRISMA checklist as Supporting Information.

---

## [Decision Letter · Decision Letter 1]

28 Jun 2022

Facing Flood Disaster: A Cluster Randomized Trial Assessing Communities' Knowledge, Skills and Preparedness utilizing a Health Model Intervention.

PONE-D-21-36005R1

Dear Dr. Kadir Shahar,

We’re pleased to inform you that your manuscript has been judged scientifically suitable for publication and will be formally accepted for publication once it meets all outstanding technical requirements.

Kind regards,

Beverley J Shea, Ph.D

Academic Editor

PLOS ONE

Additional Editor Comments (optional):

Reviewers' comments:

Reviewer's Responses to Questions

**Comments to the Author**

1. If the authors have adequately addressed your comments raised in a previous round of review and you feel that this manuscript is now acceptable for publication, you may indicate that here to bypass the “Comments to the Author” section, enter your conflict of interest statement in the “Confidential to Editor” section, and submit your "Accept" recommendation.

Reviewer #1: All comments have been addressed

Reviewer #2: All comments have been addressed

2. Is the manuscript technically sound, and do the data support the conclusions?

Reviewer #1: Yes

Reviewer #2: Yes

3. Has the statistical analysis been performed appropriately and rigorously? 

Reviewer #1: Yes

Reviewer #2: I Don't Know

4. Have the authors made all data underlying the findings in their manuscript fully available?

Reviewer #1: Yes

Reviewer #2: Yes

5. Is the manuscript presented in an intelligible fashion and written in standard English?

Reviewer #1: Yes

Reviewer #2: Yes

6. Review Comments to the Author

Reviewer #1: (No Response)

Reviewer #2: (No Response)

7. PLOS authors have the option to publish the peer review history of their article (what does this mean?). If published, this will include your full peer review and any attached files.

Reviewer #1: No

Reviewer #2: No

---

## [Editor Report · Acceptance letter]

4 Aug 2022

PONE-D-21-36005R1 

Facing Flood Disaster: A Cluster Randomized Trial Assessing Communities' Knowledge, Skills and Preparedness utilizing a Health Model Intervention. 

Dear Dr. Kadir Shahar:

I'm pleased to inform you that your manuscript has been deemed suitable for publication in PLOS ONE. Congratulations! Your manuscript is now with our production department. 

Kind regards, 

on behalf of

Dr. Beverley J Shea 

Academic Editor

PLOS ONE